CUTTING EDGE

# Promoting international collaboration and creativity in doctoral students

**Abstract** Staff from the Mayo Clinic in the US and the Karolinska Institute in Sweden describe a joint transatlantic course intended to broaden the horizons of the next generation of researchers in the field of regenerative medicine.

CHRISTOPHER M GROEN, CORMAC MCGRATH, KATHERINE A CAMPBELL, CECILIA GÖTHERSTRÖM, ANTHONY J WINDEBANK AND NATALIA LANDÁZURI*

*For correspondence: natalia.
landazuri@ki.se

Reviewing editor: Emma
Pewsey, eLife, United Kingdom

## Introduction

Modern biomedical research seeks to revolutionize clinical care through the development of new ways to combat tissue damage and disease. To contribute to this effort a researcher must understand a broad range of biomedical concepts and also have access to expertise in specific areas of research. So how might a researcher at the start of his or her career go about forming the interdisciplinary and collaborative networks that are needed to thrive in this environment, especially if the relevant expertise cannot be found in their own institution? Here we describe the design, implementation and evaluation of a joint transatlantic course in regenerative medicine that sought to prepare doctoral students for such challenges. The course was developed by employing a competency-based approach (*Frank et al., 2010*; *Vaitsis et al., 2016*) to learning design, together with elements of peer learning (*Boud and Lee, 2005*), and relied heavily on videoconferencing and other digital technology. Based on our own reflections and the feedback we received from the students at both institutions – the Mayo Clinic (MC) in Rochester, Minnesota, and the Karolinska Institutet (KI) in Stockholm – we also discuss the benefits we witnessed and the challenges we faced from a technological and a pedagogical perspective.

## Designing a joint course to promote creative thinking

Building upon a 2015 project to test the viability of holding joint, videoconference-assisted lectures, we revised our course design for 2016 with the purpose of stimulating creative thinking through collaborative work. The new design of the course offered synchronous and asynchronous elements (*Hrastinski, 2008*) over a span of 11 weeks, with each week devoted to a particular topic within regenerative medicine (for example, cardiovascular regenerative medicine, central nervous system regeneration, and ethics and policies). The course structure, described in detail below, consisted of: off-class online work; in-class learning sessions; and an examination task.

### Off-class online work

In between weekly in-class sessions, we assigned the students to read a scientific article related to the topic of the upcoming lecture, which was suggested and often authored by the upcoming speaker, an expert in the field. To stimulate creative thinking and collaboration, we asked the students to share their reflections around the topic of the article using the online learning management system (LMS) Blackboard. Students had to critically analyze the article, and propose innovative hypotheses, methods or projects that would derive from their

**Figure 1.** The components and strategies used in the course "Regenerative Medicine: Principles to Practice". (A) Schematic illustration of off-class online work and in-class learning sessions. (B) Learning strategies used in the course design.

learning and their research experience as doctoral students. After students had submitted their own analysis, they were privy to the work of others. We encouraged them to add comments and start a discussion online. To further support inter-institutional collaboration, two students, one from each institute, teamed up to assemble all the reflections and act as journal club leaders during the upcoming in-class sessions (*Figure 1A*).

### In-class learning session

Students, moderators and speakers synchronously shared the in-class session through videoconferencing, actively interacting during a two-hour session. During the first hour, the assigned journal club leaders led the discussion, where all students participated actively, usually without the speaker's involvement. As a continuation, the speaker, a lead researcher from KI or MC (alternating weekly), presented his or her latest

findings for the span of 30 minutes. Given the informal nature of the lecture, speakers often shared unpublished research findings. Finally, students actively engaged in a discussion with the speaker for an additional 30 minutes (*Figure 1A*).

### Examination

The examination involved writing a hypothesis-driven project proposal related to the field of regenerative medicine. Students were asked to form groups of 2–3 students, based on complementary interests and expertise, that included members from both KI and MC. Each group was challenged with the task of proposing a novel, to date unexplored, concept. In addition to addressing the background, hypothesis, methods and significance of the proposal, we asked the students to clearly explain the aspects of the proposal they considered innovative and describe how each collaborator contributed to building the proposal. We then assigned the students to peer review another group's proposal and revise their own proposal based on the feedback. Students at MC were tasked to do a presentation and defense of their project proposal, where students were given the opportunity to learn from and question their peers' proposals. Due to scheduling conflicts, students from KI did not participate in the oral defense of the research proposal, but participated fully in the written submission and the peer review. In recognition of the educational value of the oral presentation, we have revised the schedule to include synchronous joint presentations in the next iteration of our course in the spring of 2017.

At the end of the course, we found that students considered all aspects of the course as drivers of creative thinking, in particular their interactions with experts in the field. When asked to rank the course components, the students attributed the highest scores to the scientific presentations, interactions with the speakers and the final examination (*Figure 2A*). In agreement with this, the proposed projects included unconventional and state-of-the-art concepts from the weekly lectures. The proposals clearly showed creative and critical thinking, as well as hypothetical concepts that had been developed well beyond individual comments from online reflections or discussions.

## Logistics and practicalities

The preparatory phase was crucial for implementing the joint course because we had to carefully examine the intricacies of pre-established course formats within each institution. For example, MC utilizes a letter grading scale while KI utilizes a pass/fail system. At MC, standard courses during the spring semester last 12 weeks. At KI the duration and span is flexible. While we could not fully reconcile all the differences, we made sure to explain them to the students. We defined our schedule for the synchronous in-class sessions within the confinements of time zones (a 7-hour difference between institutions), with students from MC attending the sessions from 8 to 10 am, and the students from KI from 3 to 5 pm. Finally, at MC, teaching assistants, often doctoral students or postdoctoral fellows, are instrumental in the practical and daily coordination of teaching and assessment activities. At KI, course organizers (mostly assistant professors) are responsible for these roles. We found it crucial to establish a constant line of communication between the teaching assistant from MC and one assigned course leader from KI to ensure that all students received the same timely access to resources and information.

From a technological perspective, we had to secure access to facilities with stable videoconference systems and utilize a functional online platform to facilitate communication outside the classroom. We used the Polycom and the Cisco videoconference systems at KI and MC respectively. This allowed bidirectional transmission of video, voice and content. Although these systems offer high stability, we acknowledge that the success of videoconferencing depended on careful prior testing of the system and on access to technical support during the sessions. Thanks to technical support, delays were minor and did not have a perceptible negative impact on the planned activities. In fact, the vast majority of students (92%) viewed videoconferencing as an effective tool to hold joint sessions between MC and KI. They perceived that the scientific presentations and the student-speaker interactions had virtually the same high quality regardless of whether this presentation or interaction was in person or through the screen (*Figure 3*). They also ranked highly their level of interaction with students from the partner institution, even though they found it somewhat easier to interact with peers from their own institution (*Figure 3*). Informal spontaneous interactions are probably more likely to occur when students share a physical location, for example immediately before or after an in-class session, while

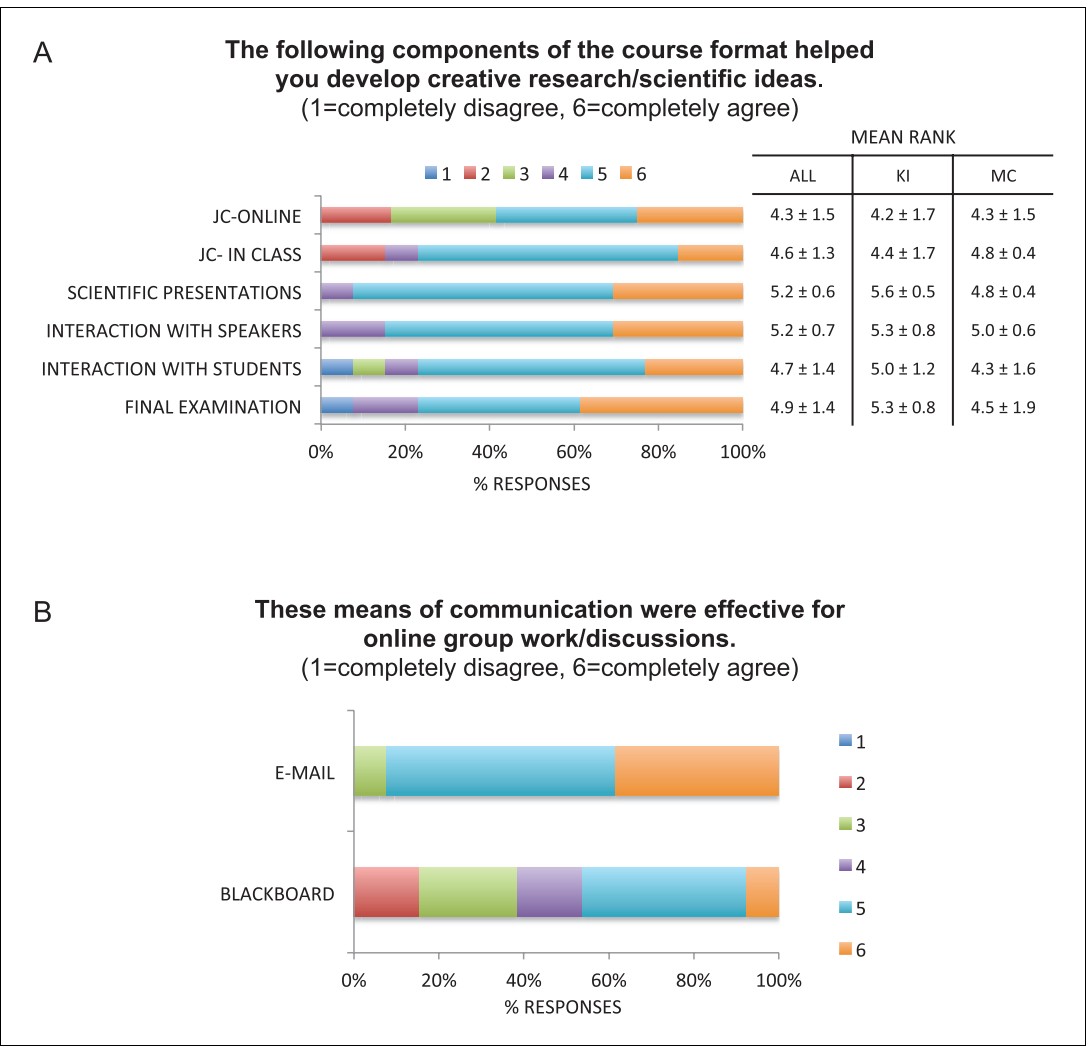

**Figure 2.** Student evaluations of the effectiveness of the collaborative course. (**A**) Students ranked the extent to which course components helped them develop creative research/scientific ideas on a scale from 1 to 6 (1=lowest, 6=highest). Mean rankings from students from both institutions (ALL, 13 respondents), from KI (7 respondents), and from MC (6 respondents) are reported as mean ± standard deviation. JC: journal club. (**B**) Students ranked the extent to which email and Blackboard were effective for student-student interactions (1=lowest, 6=highest).

interactions with peers from the partner institution require more structure and planning.

Online tools were valuable to mediate interactions between members of both institutions outside the classroom. We chose to use the Blackboard LMS to provide written information to the students, as it offers excellent tools for course organization, communication, and dissemination of information (*Bradford et al., 2007*). It can also promote learning through asynchronous

student-student interactions and discussions (*Green et al., 2014*). We asked the students to use Blackboard to post their journal club reflections and discussions. MC uses this platform and was able to incorporate course leaders and students from KI as users. Interestingly, when deciding how to interact to co-lead journal club and to prepare and execute the final examination, students preferred to communicate via email (*Figure 2B*) or use other tools they were likely familiar with, such as iMessage, Skype, Facetime, Google Docs and Slides. In fact, others have also reported that LMS are not the students' preferred means of communication (*Back et al., 2016*). In our experience, a dedicated LMS was highly beneficial as a repository for course information.

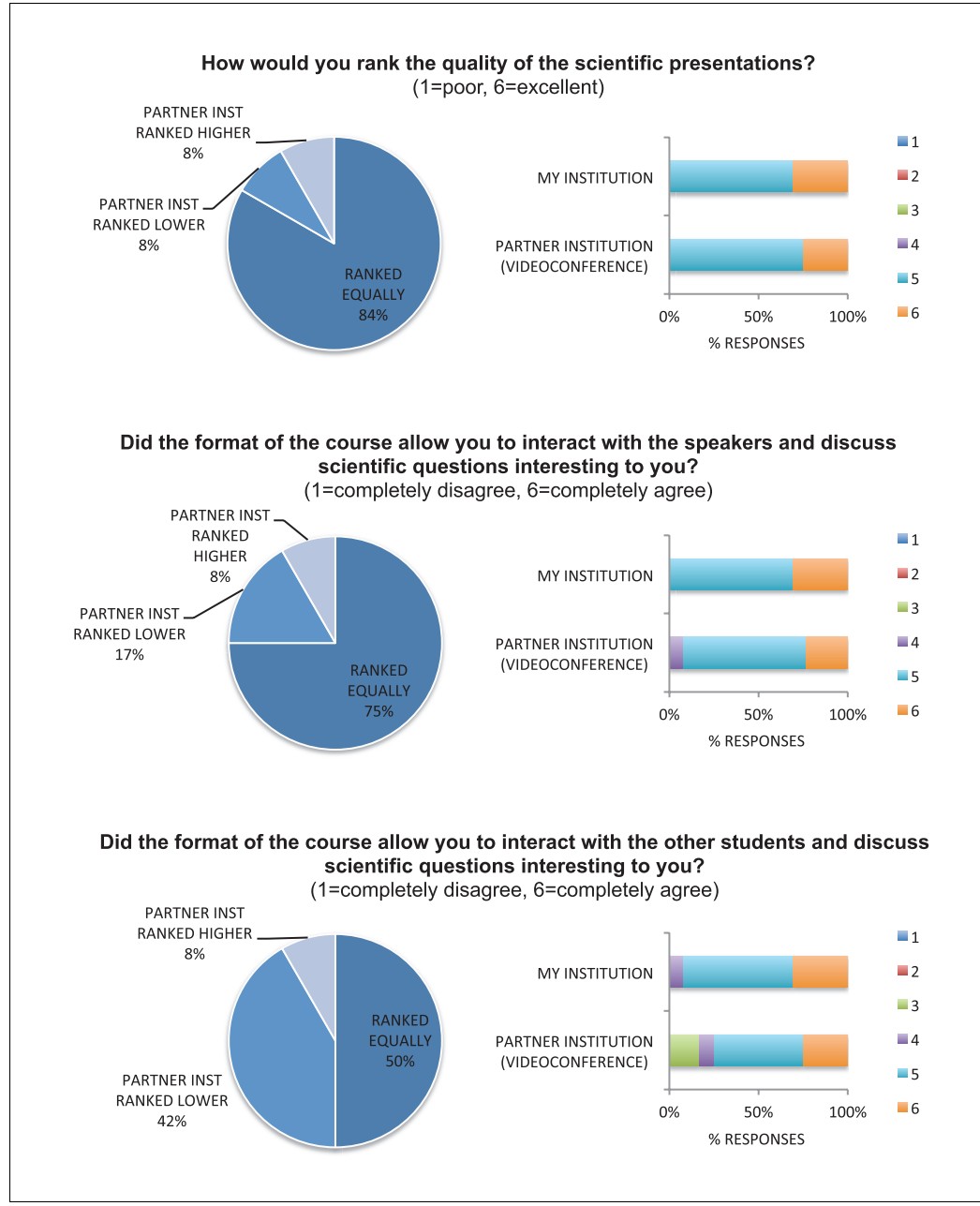

**Figure 3.** Student evaluations of the quality of the collaborative course. Students ranked the quality of scientific presentations, as well as their level of interaction with speakers and students, on a scale from 1 to 6 (1=lowest, 6=highest). Rankings for their own institution and for their partner institution are presented separately.

## Implementing a transatlantic course

We found the experience of implementing this course highly rewarding for both students and course organizers. Students and instructors alike had the opportunity to learn from experts in an informal, relaxed learning environment, and the students had the opportunity to team up and develop creative proposals that aim to tackle complex biomedical problems using state-of-the-art knowledge. We made this possible through establishing a robust technology-assisted environment. Videoconferencing and online tools were effective at mediating synchronous and asynchronous interaction between members of both institutions. In addition, the course design motivated students to take a creative and collaborative approach. We

incorporated many pedagogical strategies that have been shown to enhance student learning, including blended learning environments – online learning systems and face-to-face instruction (*Robin et al., 2011*; *Ruiz et al., 2006*; *Banerjee, 2011*), collaborative learning, and active student involvement (*Al Achkar, 2016*; *Connell et al., 2016*), including elements of peer learning (*Boud and Lee, 2005*) (*Figure 1B*). In doing so, we are able to explore the possibilities of broadening the scope of the educational environment. The peer-led sessions allowed students to use their networks of learning (*Boud and Lee, 2005*); the outcome-based design (*Vaitsis et al., 2016*) of the course made both asynchronous and synchronous activities possible (*Hrastinski, 2008*).

While similar efforts have been performed previously, we argue that our course constitutes an important step toward training students to develop professional collaborations within an international research network and independent of their principal investigator in the context of regenerative medicine. This type of training is of particular importance for doctoral students interested in pursuing an academic research career. Thus, we propose that new joint courses could explore the possibility of building actual grant proposals, or even executing short but innovative collaborative projects. Here, we acknowledge a need to increase and enhance the feedback students give to each other during unmonitored asynchronous learning (*Hrastinski, 2008*; *Boud and Molloy, 2013*).

We conclude with some practice points for institutions interested in organizing a course similar to ours.

### Practice points

- Define common grounds for learning outcomes, learning activities and examinations.
- Identify and prioritize important aspects of the course that require the active participation of students from both sides of the screen during synchronous sessions.
- Keep an even balance between the number of students from both collaborating institutions.
- Course facilitators need to be ready to solve technical issues or to contact technical support. They should ensure a balanced interaction between members from both sides of the screen but should otherwise not dominate the conversation.
- The formation of groups to design inter-institutional collaborative projects can be self-propelled or facilitated by the course organizers. In either scenario, members within each group should have complementary interests and ideas.

## Acknowledgements

We thank the Regenerative Medicine and the Development and Regeneration Doctoral Programs at Karolinska Institutet for providing funding and support for the implementation of this course. We thank the Clinical and Translational Sciences Predoctoral Program, the Center for Regenerative Medicine and the Department of Molecular Pharmacology and Experimental Therapeutics at Mayo Clinic. This project was partly supported by CTSA Grant Number TL1 TR000137 from the National Center for Advancing Translational Science (NCATS). Its contents are solely the responsibility of the authors and do not necessarily represent the official views of the NIH. We thank Cindy Gosse and Harold (Tony) Hanson for course organization and technical support at Mayo Clinic. We also thank the Mayo Clinic Media Support Services Department for successful implementation of videoconferencing. We thank Evren Alici, Pauliina Damdimopoulou, Fredrik Lanner and Hong Qian, for contributing as course organizers and facilitators at Karolinska Institutet. We also thank technical support from Karolinska Institutet.

**Christopher M Groen** is at the Center for Regenerative Medicine, Mayo Clinic, Rochester, United States

**Cormac McGrath** is in the Department of Learning, Informatics, Management and Ethics, Karolinska Institutet, Stockholm, Sweden

http://orcid.org/0000-0002-8215-3646

**Katherine A Campbell** is in the Department of Molecular Pharmacology and Experimental Therapeutics, Mayo Clinic, Rochester, United States

**Cecilia Götherström** is in the Department of Clinical Science Intervention and Technology and the Center for Hematology and Regenerative Medicine, Karolinska Institutet, Stockholm, Sweden

http://orcid.org/0000-0003-3782-0048

**Anthony J Windebank** is at the Center for Regenerative Medicine, Mayo Clinic, Rochester, United States

**Natalia Landázuri** is in the Department of Medicine, Karolinska Institutet, Stockholm, Sweden

http://orcid.org/0000-0002-7182-8767

*Competing interests:* The authors declare that no competing interests exist.

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
