## [Decision Letter]

[Editors’ note: the article was further edited in response to editorial feedback after the revised version was submitted, so some of the authors’ responses may no longer apply.]

Thank you for submitting your article “Just a click away: Using digital spaces to promote trans-Atlantic collaboration and creativity in doctorial students” to *eLife* for consideration as a Feature Article. Your article has been reviewed by two peer reviewers. I (Reviewing Editor) have overseen the evaluation and have drafted this decision letter based on the reviewers' reports to help you prepare a revised submission.

The following individuals involved in the review of your submission have agreed to reveal their identity: Derek Groen (Reviewer #1).

Summary:

The reviewers read your paper with pleasure and appreciated the arguments that doctoral students in regenerative medicine require access to scientific understanding of interdisciplinary biomedical concepts and to novel technological solutions. The description of how a collaborative technology-assisted platform has assisted student learning and the creation of research networks is one that will be of value and interest to a broad audience.

Essential revisions:

1) In terms of presentation, the paper mixes the “creation of research networks” with “joint transatlantic teaching” in a manner that is confusing to the reader. The story could be sharpened by a clarification along the following lines: by collaborative transatlantic teaching & scholarship (through the student-led journal clubs), students become familiarized with the mechanisms involving in transatlantic collaborations and help prepare them for the future participation in international research networks.

2) In relation to point 1, it may be a bridge too far to claim that this approach stimulates “the creation of research networks”, as those networks have been largely pre-constructed as part of the module design.

3) The figure is of benefit to the paper, but a second one would help clarify things much further. Such a second figure should clearly contain the teaching elements and techniques that made up this distributed module, so that the reader can see what existing practices have been combined to create this unique new way of teaching (subsection “Implementing a successful translatlantic doctoral course”). The figure doesn't have to be exhaustive, but it should serve as a strong vehicle to clearly convey the key elements needed for this, and give the reader an idea of what they need to apply the digital spaces-based teaching in their own academic context.

4) The subsections of “Designing a joint course to promote collaborative creative thinking” do not fit well with the categorization at the start of that page (student-led journal club, guest lecture, and discussion session). Please revise those subsections, so that they match well either with the categorization at the start, or with the elements of your “new” Figure 2.

5) Please revise the descriptions in the “Examination” subsection. Especially, did the students of KI not participate only in the presentation/defense part, or were they excluded from the examination altogether? You should also avoid writing “we regret this decision” when a scheduling conflict clearly beyond your control has prevented you from doing this particular activity.

6) Please provide the data from your surveys that supports the statements made about the students’ experiences of the course. Below is a suggestion for how you could incorporate some of this information into the article, based on the PDF of survey results that you sent me when we initially discussed writing a Feature article.

– The top figure on page 5 of your survey results PDF (with the percentage of responses) could be added to the article as a new figure; the results from the bottom figure on page 5 (with the mean rank) could be integrated into the top figure (for example, by moving the explanation of the colours to above or below the figure, and printing the mean rank for each row to the right of that row). Table 1 could then be deleted.

– The figure on page 1 of the results PDF (pie chart with the percentage of responses on the effectiveness of videoconferencing) contains just two data points so these could be worked into the text and/or the caption for the new figure.

– The figures on page 2/3/4 of the results PDF (from the results of the quality of scientific presentations to the effectiveness of means of communication for online group work/discussions) could become figure supplements to the new figure. This might involve deleting Table 2.

7) Further to point 6, please provide the number of students on the module respectively for both institutions. It would also be good to see a breakdown of the student ratings from both MC and KI (currently shown in Table 1). Please also provide standard deviation values (perhaps using the commonly adopted +- notation for ease of reading) to give an idea of the uncertainty range.

8) It would be good to comment on the use of “reciprocal peer learning”, given that the students are interacting on various platforms. And if you do, you should perhaps add a comment or two on the benefits of peer learning to student learning (see for example articles by David Boud), and also on whether this type of learning has already been successfully used in online environments. In addition, what is the evidence for “international” interaction using these learning strategies? Is this study the first of its kind? A review of literature would suggest otherwise.

---

## [Author Response]

*Essential revisions:*

*1) In terms of presentation, the paper mixes the “creation of research networks” with “joint transatlantic teaching” in a manner that is confusing to the reader. The story could be sharpened by a clarification along the following lines: by collaborative transatlantic teaching & scholarship (through the student-led journal clubs), students become familiarized with the mechanisms involving in transatlantic collaborations and help prepare them for the future participation in international research networks.*

We recognize the importance of this clarification and have included the following statement within the Introduction to help distinguish between the collaborative teaching/scholarship and the actual creation of research networks in the future.

The additional sentence reads, “Through this model of collaborative teaching and scholarship, it is our aim that students become familiarized with mechanisms involved in transatlantic collaboration in order to prepare them for future participation in international research networks.”.

*2) In relation to point 1, it may be a bridge too far to claim that this approach stimulates “the creation of research networks”, as those networks have been largely pre-constructed as part of the module design.*

We agree with this comment and have revised the conclusions of this paper accordingly. In order to emphasize the difference between training students in the mechanisms of international collaboration and the actual creation of research networks, the last paragraph of the manuscript discussion now reads, “Given our experiences outlined above, we would like to encourage other international institutions to implement a design similar to ours in the interest of sharing scientific resources and training early-career researchers to create international research networks.”.

*3) The figure is of benefit to the paper, but a second one would help clarify things much further. Such a second figure should clearly contain the teaching elements and techniques that made up this distributed module, so that the reader can see what existing practices have been combined to create this unique new way of teaching (subsection “Implementing a successful translatlantic doctoral course”). The figure doesn't have to be exhaustive, but it should serve as a strong vehicle to clearly convey the key elements needed for this, and give the reader an idea of what they need to apply the digital spaces-based teaching in their own academic context.*

We thank the reviewers for the suggestion, and agree that a second figure will improve the paper. We have created a new figure (Figure 5) that shows the teaching techniques that were used in this course, and how the various course elements fit into our teaching model.

4) The subsections of “Designing a joint course to promote collaborative creative thinking” do not fit well with the categorization at the start of that page (student-led journal club, guest lecture, and discussion session). Please revise those subsections, so that they match well either with the categorization at the start, or with the elements of your “new” Figure 2.

We thank the reviewers for the suggestion and have modified the section to include a clearer categorization of the course structure. The introductory paragraph now states the three major course components: (A) off-class online work, (B) in-class learning occasions, and (C) an examination task. The subsequent three paragraphs use these as subsection titles for the in depth discussion of each component.

*5) Please revise the descriptions in the “Examination” subsection. Especially, did the students of KI not participate only in the presentation/defense part, or were they excluded from the examination altogether? You should also avoid writing “we regret this decision” when a scheduling conflict clearly beyond your control has prevented you from doing this particular activity.*

In order to avoid confusion about the KI students’ involvement in the final course examination, we have revised the manuscript to read, “Due to scheduling conflicts, students from KI did not participate in the oral defense of the research proposal, but participated fully in the written submission and the peer review. In recognition of the educational value of the oral presentation, we have revised the schedule to include synchronous joint presentations in the next iteration of our course in the spring of 2017.”

*6) Please provide the data from your surveys that supports the statements made about the students’ experiences of the course. Below is a suggestion for how you could incorporate some of this information into the article, based on the PDF of survey results that you sent me when we initially discussed writing a Feature article.*

*– The top figure on page 5 of your survey results PDF (with the percentage of responses) could be added to the article as a new figure; the results from the bottom figure on page 5 (with the mean rank) could be integrated into the top figure (for example, by moving the explanation of the colours to above or below the figure, and printing the mean rank for each row to the right of that row). Table 1 could then be deleted.*

We have followed the reviewer’s suggestion and have included the top figure on page 5 of our survey results and added the mean rank for each component to the right of the figure. This corresponds to Figure 2 in our revised article. Also, we have deleted Table 1.

– The figure on page 1 of the results PDF (pie chart with the percentage of responses on the effectiveness of videoconferencing) contains just two data points so these could be worked into the text and/or the caption for the new figure.

These data points are included in the text: “In fact, the vast majority of students (92%) viewed videoconferencing as an effective tool to hold joint sessions between MC and KI”.

– The figures on page 2/3/4 of the results PDF (from the results of the quality of scientific presentations to the effectiveness of means of communication for online group work/discussions) could become figure supplements to the new figure. This might involve deleting Table 2.

We have incorporated these survey results as new figures in the article: Figures 3 and 4. We have removed Table 2.

*7) Further to point 6, please provide the number of students on the module respectively for both institutions. It would also be good to see a breakdown of the student ratings from both MC and KI (currently shown in Table 1). Please also provide standard deviation values (perhaps using the commonly adopted +- notation for ease of reading) to give an idea of the uncertainty range.*

The total number of students that took the course in 2016 was 17. Thirteen of those students chose to participate in the voluntary anonymous survey we conducted. We have specified the number of respondents from each institution in the legend of Figure 2. We have also added a breakdown or the students’ ratings for both MC and KI, including standard deviation values, as part of Figure 2.

*8) It would be good to comment on the use of “reciprocal peer learning”, given that the students are interacting on various platforms. And if you do, you should perhaps add a comment or two on the benefits of peer learning to student learning (see for example articles by David Boud), and also on whether this type of learning has already been successfully used in online environments.*

We acknowledge and appreciate the reference to Boud. Boud’s extensive work on peer learning has been inspirational to us and instrumental in some ways. It was not added to the first draft as it was felt that there was too little space to elaborate on the concept of peer learning in the context of the submission format. It is now added as a reference in the initial description of the design of the course. Further, we have tried to acknowledge the potential of using peer learning in the context of doctoral education: This is done in particular in the subsection “Implementing a transatlantic course”:

“…[i]ncluding elements of peer learning (Boud and Lee, 2005) (Figure 5). […] Here, we acknowledge a need to increase and enhance the feedback students give to each other in the context of the unmonitored asynchronous learning (Hrastinski, 2008; Boud and Molloy, 2013).”

In addition, what is the evidence for “international” interaction using these learning strategies? Is this study the first of its kind? A review of literature would suggest otherwise.

We have tried to acknowledge some of the previous work done on identifying the specific transatlantic and web-based dimensions of doctoral education. Here we have also endeavored to acknowledge that our efforts are an important step in the context of regenerative medicine specifically. This is done in subsection “Implementing a transatlantic course”:

“While similar efforts have been done previously, we argue that our course constitutes an important step toward training students to develop professional collaborations within an international research network and independent of their principal investigator in the context of regenerative medicine. “